# Read Between the Layers: Leveraging Multi-Layer Representations for Rehearsal-Free Continual Learning with Pre-Trained Models

**Kyra Ahrens**[*]                                                  *kyra.ahrens@uni-hamburg.de*
*University of Hamburg*

**Hans Hergen Lehmann**[*]                          *hergen.lehmann@studium.uni-hamburg.de*
*University of Hamburg*

**Jae Hee Lee**                                              *jae.hee.lee@uni-hamburg.de*
*University of Hamburg*

**Stefan Wermter**                                      *stefan.wermter@uni-hamburg.de*
*University of Hamburg*

**Reviewed on OpenReview:** *https://openreview.net/forum?id=ZTcxp9xYr2*

## Abstract

We address the Continual Learning (CL) problem, wherein a model must learn a sequence of tasks from non-stationary distributions while preserving prior knowledge upon encountering new experiences. With the advancement of foundation models, CL research has pivoted from the initial learning-from-scratch paradigm towards utilizing generic features from large-scale pre-training. However, existing approaches to CL with pre-trained models primarily focus on separating class-specific features from the final representation layer and neglect the potential of intermediate representations to capture low- and mid-level features, which are more invariant to domain shifts. In this work, we propose LayUP, a new prototype-based approach to CL that leverages second-order feature statistics from multiple intermediate layers of a pre-trained network. Our method is conceptually simple, does not require access to prior data, and works out of the box with any foundation model. LayUP surpasses the state of the art in four of the seven class-incremental learning benchmarks, all three domain-incremental learning benchmarks and in six of the seven online continual learning benchmarks, while significantly reducing memory and computational requirements compared to existing baselines. Our results demonstrate that fully exhausting the representational capacities of pre-trained models in CL goes well beyond their final embeddings.

## 1 Introduction

Continual Learning (CL) is a subfield of machine learning dedicated to developing models capable of learning from a stream of data while striking the balance between adapting to new concepts and retaining previously acquired knowledge without forgetting (Parisi et al., 2019; Chen & Liu, 2018). While traditional works on CL primarily focus on the learning-from-scratch paradigm, the introduction of large foundation models has initiated a growing interest in developing CL methods upon the powerful representations resulting from large-scale pre-training. Continual learning with pre-trained models builds on the assumption that a large all-purpose feature extractor provides strong knowledge transfer capabilities and great robustness to catastrophic forgetting (McCloskey & Cohen, 1989) during incremental adaptation to downstream tasks.

---

[*]Equal contribution.

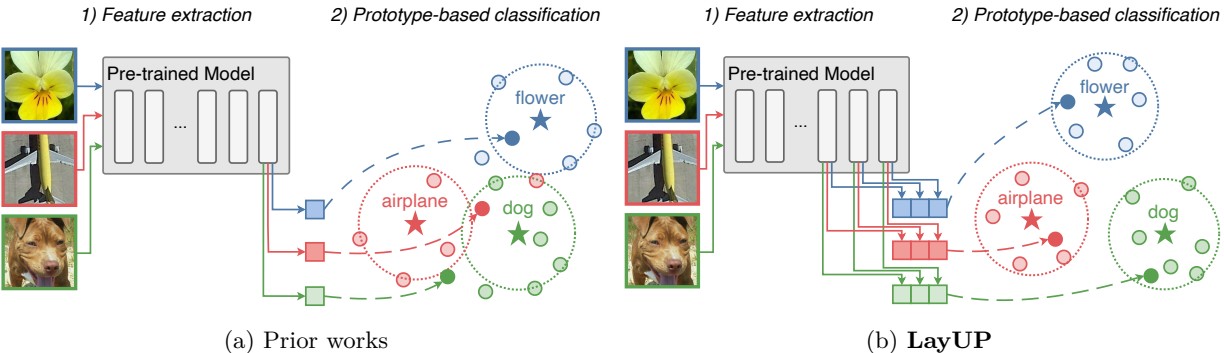

(a) Prior works                                           (b) **LayUP**

Figure 1: **Overview of LayUP.** Prior works on CL with pre-trained models use only final representation layer features for classification. LayUP enhances the last layer representations of pre-trained models by incorporating features from intermediate layers, resulting in more accurately calibrated similarity measures for prototype-based classification and greater robustness to domain gaps in the CL setting. Stars (⋆) denote class prototypes.

Recent approaches to CL with pre-trained models primarily adopt three main strategies (Wang et al., 2024; McDonnell et al., 2023): (i) carefully fine-tuning the parameters of the pre-trained model (*full body adaptation*), (ii) keeping the pre-trained parameters fixed while learning a small set of additional parameters such as prompts (Lester et al., 2021; Liu et al., 2023; Jia et al., 2022), adapters (Rebuffi et al., 2017a; Houlsby et al., 2019; Hu et al., 2022; Chen et al., 2022; 2023), or others (Ke et al., 2021; Marullo et al., 2023; Lian et al., 2022), or (iii) extracting class prototypes from pre-trained representations without further fine-tuning. All three strategies share the beneficial property of not relying on rehearsal from past data to perform well, unlike many of the top-performing CL strategies used for models trained from scratch. The optimal strategy for balancing the trade-offs between effectively utilizing pre-trained features, ensuring resource efficiency, and maintaining robustness to domain shifts in CL settings remains an open question.

Full body adaptation, as done with strategy (i), is expensive with respect to computational resources and required training data. Parameter-Efficient Transfer Learning (PETL), as implemented in strategy (ii), updates a fully connected classification head (*linear probe*) via iterative gradient methods during training and inference, which is subject to task-recency bias (Mai et al., 2021; Rymarczyk et al., 2023; Wang et al., 2023) and catastrophic forgetting (Zhang et al., 2023; Ramasesh et al., 2021). Class-prototype methods (Zhou et al., 2023b; Janson et al., 2022; McDonnell et al., 2023) that follow strategy (iii), in contrast, offer a promising alternative, as they utilize extracted features directly, making them more resource-efficient than full body adaptation and demonstrating greater robustness compared to PETL methods applied to the trainable linear probe.

Despite their effectiveness, contemporary class-prototype methods for CL extract only the features obtained from the last layer of the backbone for prototype construction. However, as the discrepancy between the pre-training and fine-tuning domains widens, the high-level features from the last layer may not generalize sufficiently to ensure adequate class separability in the target domain. Thus, the challenge lies in developing a prototype-based classifier that utilizes the representations of a pre-trained feature extractor in a holistic manner, ideally with low memory and computational costs.

In this work, we hypothesize that representations are most effectively leveraged from multiple layers to construct a classifier based on first-order (class prototypes) and second-order (Gram matrix) feature statistics. Such a strategy is inspired by works on Neural Style Transfer (Gatys et al., 2016; Jing et al., 2020), which involve applying the artistic style of one image to the content of another image, a problem that can be viewed through the lens of domain adaptation (Li et al., 2017). To achieve neural style transfer, a model has to disentangle image information into content and style (e.g., textures, patterns, colors). While content information is expressed in terms of activations of features at different layers (which translates to calculating multi-layer class prototypes in our case), style information is expressed as correlations between features of different layers (which corresponds to calculating multi-layer Gram matrices).

We thus propose Multi-**Lay**er **U**niversal **P**rototypes (**LayUP**), a class-prototype method for CL that is based on the aforementioned strategy to disentangle style and content information of images at multiple layers of a pre-trained network. An overview of our method is given in Fig. 1. LayUP obtains rich and fine-grained information about images to perform a more informed classification that is less sensitive to domain shifts. We additionally experiment with different parameter-efficient fine-tuning strategies to further refine the intermediate representations obtained by LayUP. Across various CL benchmarks and learning settings, our method enhances performance and narrows the gap to the upper bound by as much as 80% (Stanford Cars-196 in the CIL setting, cf. Sec. 5.2) compared to the next best baseline, while also reducing its memory and compute requirements by 81% and up to 90% (cf. Sec. 4.3), respectively. Beyond that, our method can serve as a versatile plug-in to enhance existing class-prototype methods, consistently yielding an absolute performance gain of up to 31.1% (cf. Sec. 5.7).

Our contributions are threefold: (1) We present and examine in detail the CL strategy with pre-trained models and show why it benefits from extracting intermediate representations directly for classification. We further demonstrate the advantages of leveraging cross-correlations of features within and between multiple layers to decorrelate class prototypes. (2) Building on our insights, we propose LayUP, a novel class-prototype approach to CL that leverages second-order statistics of features from multiple layers of a pre-trained model. Inspired by prior works (Zhou et al., 2023a; McDonnell et al., 2023), we experiment with different methods for parameter-efficient model tuning and extend our method towards first session adaptation. Our final approach is not only conceptually simple, but also benefits from low memory and computational requirements, and integrates seamlessly with any pre-trained model. (3) We report performance improvements with a pre-trained ViT-B/16 (Dosovitskiy et al., 2021) backbone on the majority of benchmarks the Class-Incremental Learning (CIL) and Domain-Incremental Learning (DIL) settings (Van De Ven et al., 2022) as well as in the challenging Online Continual Learning (OCL) setting. We show that LayUP is especially effective under large distributional shifts and in the low-data regime. Our results highlight the importance of taking a deeper look at the intermediate layers of pre-trained models to better leverage their powerful representations, thus identifying promising directions for CL in the era of large foundation models. The source code is available at https://github.com/ky-ah/LayUP.

## 2 Related Work

### 2.1 Continual learning

Traditional approaches to CL consider training a model from scratch on a sequence of tasks, while striking a balance between adapting to the currently seen task and maintaining high performance on previous tasks. They can be broadly categorized into the paradigms of *regularization* (Kirkpatrick et al., 2017; Zenke et al., 2017; Li & Hoiem, 2018; Aljundi et al., 2018; Dhar et al., 2019), which selectively restricts parameter updates, *(pseudo-)rehearsal* (Rebuffi et al., 2017b; Chaudhry et al., 2019; Buzzega et al., 2020; Prabhu et al., 2020), which recovers data either from a memory buffer or from a generative model, and *dynamic architectures* (Rusu et al., 2016; Yoon et al., 2018; Serra et al., 2018), which allocate fully or partially disjoint parameter spaces for each task.

### 2.2 Continual learning with pre-trained models

Leveraging the powerful representations from large foundation models for downstream continual learning has shown to not only facilitate knowledge transfer, but also to increase robustness against forgetting (Ramasesh et al., 2022; Ostapenko et al., 2022). L2P (Wang et al., 2022c) and DualPrompt (Wang et al., 2022b) outperform traditional CL baselines by training a pool of learnable parameters (i.e., prompts), which serve as queries to guide a Vision Transformer (ViT) (Dosovitskiy et al., 2021) towards adapting to downstream tasks. These approaches provide the foundation for numerous more recent prompt learning methods, such as S-Prompt (Wang et al., 2022a), DAP (Jung et al., 2023), and CODA-Prompt (Smith et al., 2023a), which further enhance performance.

Contrary to prompt learning methods that keep the backbone parameters fixed while updating a small set of additional parameters, some parallel works explore methods for making careful adjustments to the backbone

parameters. SLCA (Zhang et al., 2023) uses a small learning rate for updates to the ViT backbone while training a linear probing layer with a larger learning rate. L2 (Smith et al., 2023b) applies regularization to the self-attention parameters of a ViT during continual fine-tuning. First Session Adaptation (FSA) (Panos et al., 2023) makes adjustments to the parameters of the backbone only during first task training to bridge the gap between pre-training and downstream CL domains.

As an alternative to the methods above that adjust the feature space via cross-entropy loss and backpropagation, constructing prototypes directly from the pre-trained features emerges as a cost-efficient alternative to overcome forgetting during CL. Janson et al. (2022) and Zhou et al. (2023b) find that a conceptually simple Nearest Mean Classifier (NMC) outperforms various prompt learning methods. ADAM (Zhou et al., 2023b) combines NMC with FSA and concatenates the features from the initial pre-trained ViT and the adapted parameters. Other methods leverage second-order feature statistics, i.e., covariance information, for class prototype accumulation: FSA+LDA (Panos et al., 2023) applies an incremental version of linear discriminant analysis to a pre-trained ResNet encoder (He et al., 2016). RanPAC (McDonnell et al., 2023) uses high-dimensional random feature projections to decorrelate class prototypes of a pre-trained ViT backbone.

All the aforementioned approaches have in common that they consider only the last layer representations (or, features) for classification. We provide a different perspective to the prototype-based approach to CL and show that intermediate representations can add valuable information to the linear transformation of class prototypes (cf. Eq. (6)), which leads to better class separability and increased robustness to large distributional shifts.

## 3 Preliminaries

### 3.1 Continual learning problem formulation

We consider a feature extractor $\phi(\cdot)$ composed of $L$ consecutive layers and a sequence of training datasets $\mathcal{D} = \{\mathcal{D}_1, \ldots, \mathcal{D}_T\}$, where the $t^{\text{th}}$ task $\mathcal{D}_t = \{(\boldsymbol{x}_{t,n}, y_{t,n})\}_{n=1}^{N_t}$ contains pairs of input samples $\boldsymbol{x}_{t,n} \in \mathcal{X}_t$ and their ground-truth labels $y_{t,n} \in \mathcal{Y}_t$. The total number of classes observed in $\mathcal{D}$ (i.e., the number of unique elements in $\mathcal{Y}$) is denoted with $C$. Given an arbitrary input sample $\boldsymbol{x} \in \mathcal{X}$, the $d_l$-dimensional encoded features from the $l^{\text{th}}$ layer of model $\phi(\cdot)$ are represented as $\phi_l(\boldsymbol{x}) \in \mathbb{R}^{d_l}$. Thus, the features from the last (or representation) layer of the model are given by $\phi_L(\boldsymbol{x}) \in \mathbb{R}^{d_L}$.

Our focus in this work is rehearsal-free CL, where historical data cannot be fetched for replay. In the *Class-Incremental Learning* (CIL) setting, label spaces are partially disjoint between tasks. Conversely, in the *Domain-Incremental Learning* (DIL) setting, label spaces are kept across tasks, but samples within a label space are subject to distributional shifts. *Online Continual Learning* (OCL) is a more stringent definition of the aforementioned settings, where each datum in the stream of tasks is observed only once. All three CL settings prohibit the model from accessing task identifiers during testing; notably, our method does not require access to task-specific information during training either.

### 3.2 Class-prototype methods for CL with pre-trained models

To leverage the powerful representations from large-scale pre-training while overcoming the limitations of full body adaptation and linear probing (cf. Sec. 1), *class-prototype* methods extract and accumulate features from the last layer of a pre-trained model to construct representatives for each class. The most straightforward class-prototype method is the Nearest Mean Classifier (NMC), which aggregates for each class $y \in \mathcal{Y}$ training sample features via averaging to obtain class prototype $\overline{\boldsymbol{c}}_y$:

$$\overline{\boldsymbol{c}}_y = \frac{1}{K} \sum_{t=1}^{T} \sum_{n=1}^{N_t} \mathbb{1}(y = y_{t,n}) \phi_L(\boldsymbol{x}_{t,n}), \tag{1}$$

with $\mathbb{1}(\cdot)$ denoting the indicator function and $K = \sum_{t=1}^{T} \sum_{n=1}^{N_t} \mathbb{1}(y = y_{t,n})$.

During inference, NMC selects for each test sample's feature vector the class with the smallest Euclidean distance (Janson et al., 2022) or highest cosine similarity (Zhou et al., 2023b) with its class prototype.

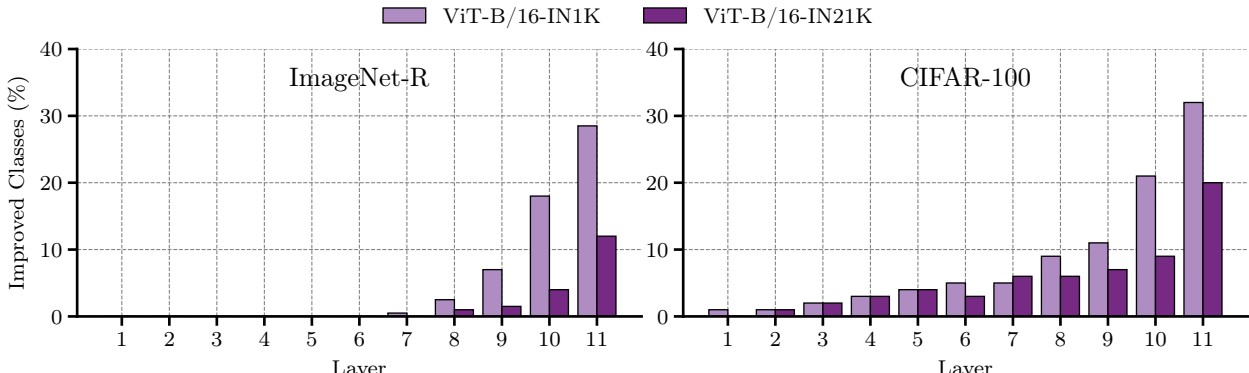

Figure 2: **Classification performance of intermediate layers.** Comparison for two different pre-training paradigms of a ViT-B/16 (Dosovitskiy et al., 2021) backbone and split ImageNet-R (*left*) and CIFAR-100 (*right*) datasets. For each intermediate layer $l \in \{1, \ldots, L-1\}$ (where $L = 12$ denotes the final representation layer), the bars represent the percentage of classes for which a classifier, utilizing Eq. (3) at layer $l$, surpasses the accuracy of the classifier at the $L^{\text{th}}$ layer.

Considering the cosine similarity measure and some $\boldsymbol{x} \in \mathcal{D}_{\text{test}}$, the predicted class label is obtained by:

$$\hat{y} = \arg\max_{y \in \{1, \ldots, C\}} s_y, \quad s_y := \frac{\phi_L(\boldsymbol{x})^T \, \overline{\boldsymbol{c}}_y}{\|\phi_L(\boldsymbol{x})\| \cdot \|\overline{\boldsymbol{c}}_y\|}, \tag{2}$$

where $\|\cdot\|$ denotes the $L^2$-norm. However, it was found that the assumption of an isotropic covariance of features (i.e., features are mutually uncorrelated) that is made under Eq. (2) does not hold for pre-trained models. To account for correlations between features as a means to better "calibrate" similarity measures, class-prototype methods that leverage second-order statistics are proposed (Panos et al., 2023; McDonnell et al., 2023). One such method is based on the closed-form ordinary least square solution to ridge regression (Murphy, 2012), which is very effective in decorrelating class prototypes in CL (McDonnell et al., 2023). It takes Gram matrix $\boldsymbol{G}$ and class prototypes (or, regressands) $\boldsymbol{c}_y$ that are aggregated via summation (instead of averaging to obtain $\overline{\boldsymbol{c}}_y$, cf. Eq. (1) with $\frac{1}{K}$ omitted) to yield:

$$\hat{y} = \arg\max_{y \in \{1, \ldots, C\}} s_y, \quad s_y := \phi_L(\boldsymbol{x})^T \, (\boldsymbol{G} + \lambda \boldsymbol{I})^{-1} \, \boldsymbol{c}_y \tag{3}$$

for a regression parameter $\lambda \geq 0$, $d_L$-dimensional identity matrix $\boldsymbol{I}$, and $\boldsymbol{G}$ expressed as summation over outer products as

$$\boldsymbol{G} = \sum_{t=1}^{T} \sum_{n=1}^{N_t} \phi_L(\boldsymbol{x}_{t,n}) \otimes \phi_L(\boldsymbol{x}_{t,n}) \tag{4}$$

Although Eq. (2) and Eq. (3) are defined with respect to the maximum number of classes $C$ after observing all $T$ tasks, they can be applied after seeing any task $t \leq T$ or any sample $n \leq N_t$. When denoting the extracted features of some input datum $\boldsymbol{x}_{t,n} \in \mathcal{D}_t$ as $\boldsymbol{f}_{t,n} = \phi_L(\boldsymbol{x}_{t,n})$ and considering $\boldsymbol{F} \in \mathbb{R}^{N \times d_L}$ as concatenated row-vector features $\boldsymbol{f}_{t,n}$ of all $N$ training samples, Eq. (4) is reduced to $\boldsymbol{G} = \boldsymbol{F}^T \boldsymbol{F}$, which corresponds to the original definition of a Gram matrix as used in the closed-form ridge estimator (Hoerl & Kennard, 1970).

## 4 LayUP

### 4.1 Class prototyping from second-order intermediate features

We argue that a combination of (i) enriching the last layer representations with hierarchical multi-layer features and (ii) decorrelating multi-layer features—which represent image properties such as content and style—via Gram matrix transformation increases robustness to domain shifts and thus improves generalizability to

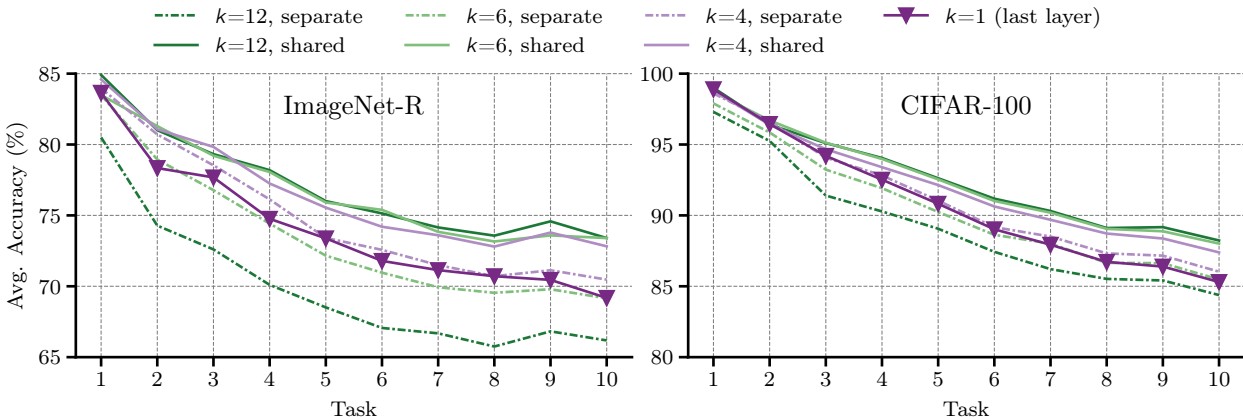

Figure 3: **Comparison of techniques to integrate intermediate representations.** LayUP implementations for different values of $k$ using a shared representation and Gram matrix as in Eq. (6) versus averaging over separate ridge (or, Gram) classifiers for each layer using Eq. (3). Results are reported as average accuracy scores over CL training on split ImageNet-R (*left*) and CIFAR-100 (*right*) datasets following phase B of Alg. 1.

downstream continual tasks. To this end, we propose to integrate embeddings from multiple layers via concatenation into a ridge regression estimator as defined in Eq. (3). Let

$$\Phi_{-k:}(\boldsymbol{x}) = (\phi_{L-k+1}(\boldsymbol{x}), \ldots, \phi_{L-1}(\boldsymbol{x}), \phi_L(\boldsymbol{x})) \tag{5}$$

denote the concatenated input features of some input sample $\boldsymbol{x} \in \mathcal{D}_{\text{test}}$, extracted from the last $k$ layers of the pre-trained model $\phi(\cdot)$. Such features can be, e.g., classification token embeddings ([CLS]) of a transformer (Vaswani et al., 2017) encoder or flattened feature maps of a ResNet (He et al., 2016) encoder. This yields a modified estimator

$$s_y := \Phi_{-k:}(\boldsymbol{x})^T \ (\boldsymbol{G} + \lambda \boldsymbol{I})^{-1} \ \boldsymbol{c}_y \tag{6}$$

with $d_{(k)} = (d_{L-k+1} + \cdots + d_{L-1} + d_L)$-dimensional identity matrix $\boldsymbol{I}$ and

$$\boldsymbol{G} = \sum_{t=1}^{T} \sum_{n=1}^{N_t} \Phi_{-k:}(\boldsymbol{x}_{t,n}) \otimes \Phi_{-k:}(\boldsymbol{x}_{t,n}) \tag{7}$$

We first motivate our approach by showing that intermediate representations capture expressive class statistics for prototype-based classification. For the split CIFAR-100 (Krizhevsky, 2009) and ImageNet-R (Hendrycks et al., 2021a) datasets and two ViT models, we construct one prototype-based classifier using Eq. (3) per layer $l \in \{1, \ldots, L-1\}$ and measure the percentage of classes that layer $l$ predicts better than the last layer $L$ ($L = 12$ for ViT-B/16). The results are provided in Fig. 2. Across both datasets, the classifiers from the last five intermediate layers outperform the representation layer classifier in up to 32% of the classes. The advantage of multi-layer classifiers is more pronounced in the ViT model pre-trained on 21 000 different ImageNet classes and additionally fine-tuned on 1 000 different ImageNet classes (IN1K) compared to the ViT model only pre-trained on 21 000 different ImageNet classes (IN21K). The results indicate that representations derived from intermediate layers generally provide meaningful knowledge to more effectively distinguish classes at various hierarchical levels.

There are two intuitive ways to integrate intermediate representations into the class-prototype classifier: Averaging over $k$ separate classifiers per Eq. (3) or concatenating representations from the last $k$ layers to obtain shared class prototypes per Eq. (6). Fig. 3 shows average accuracies for two split datasets and different values of $k$. For each $k$, transforming shared class-specific information via Gram matrix inversion, as described in Eq. (6), proves to be superior to averaging across separate classifiers. This finding indicates that the correlations across layers, captured in the shared Gram matrix (cf. Eq. (7)), add meaningful information to enhance class separability.

While performance generally benefits from higher values of $k$, the difference in performance between $k = 4$ and $k = 6$ is more pronounced than that between $k = 6$ and $k = 12$. Considering that a higher $k$ corresponds to increased computational and memory demands as $\mathbf{G}$ and $\mathbf{c}_y$ increase in dimension, this substantiates $k = 6$ as a reasonable choice for maximum representation depth. Although concatenating representations from multiple consecutive layers results in higher memory demands compared with using a final-layer classifier or averaging over layer-wise classifiers, we will demonstrate in Sec. 4.3 that our method remains significantly lighter in memory and computational requirements than other competitive class-prototype methods for CL.

Interestingly, the performance of the final-layer classifier $k = 1$, as illustrated in Fig. 3, is approximately equivalent to that achieved by averaging over separate classifiers for $k = 4$ and $k = 6$. This observation challenges the argument that class embeddings from multiple consecutive layers, which are individually strong at predicting specific classes (cf. Fig. 2), might introduce noise that impairs performance when combined via averaging.

## 4.2 Combination with parameter-efficient model adaptation

A benefit of class-prototype methods for CL is that they can be orthogonally combined with adaptation techniques to refine the pre-trained representations. As accumulated class prototypes are subject to discrepancy upon distributional shifts during supervised fine-tuning, any adaptation strategies to the latent representations should be carefully tailored to the class-prototype method used. Previous works (Panos et al., 2023; Zhou et al., 2023b) find adaptation to downstream continual tasks during First Session Adaptation (FSA) on $\mathcal{D}_1$ as sufficient to bridge the domain gap while maintaining full compatibility with CL.

To maintain the powerful generic features of the pre-trained backbone, we choose to apply FSA to an additional set of learnable PETL parameters while keeping the backbone frozen throughout. To update PETL parameters during FSA, we train a linear classification head with as many output neurons as the number of unique labels in the first task via Adam optimizer (Kingma & Ba, 2015) and cross-entropy loss only during the first CL stage and discard the classification head afterward. During the CL phase, we keep all model parameters frozen and update $\mathbf{G}$ and $\mathbf{c}_y$ only. We follow prior works (Zhou et al., 2023b; McDonnell et al., 2023) and experiment with Visual Prompt Tuning (VPT) (Jia et al., 2022), Scaling and Shifting of Features (SSF) (Lian et al., 2022), and AdaptFormer (Chen et al., 2022) as PETL methods and refer to the cited papers for details.

The pseudocode of the LayUP algorithm for the CIL setting is presented in Alg. 1. It is noteworthy that both $\mathbf{G}$ and $\mathbf{c}_{y_t}$ can be updated incrementally, one sample at a time, and Eq. (6) can be applied after every sample, thereby ensuring full compatibility with online (or, streaming) learning settings. For the OCL comparison, the FSA stage is omitted, and a fixed $\lambda$ is utilized, since a dynamic search for $\lambda$ necessitates multiple passes over the input data.

---

**Algorithm 1** LayUP Training

**Require:** Pre-trained network $\phi(\cdot)$
**Require:** PETL parameters
**Require:** Data $\mathcal{D} = \{\mathcal{D}_1, \dots, \mathcal{D}_T\}$
**Require:** $k$

  *# Initialization of class prototypes*
  $\mathbf{G} \leftarrow \mathbf{0} \in \mathbb{R}^{d_{(k)} \times d_{(k)}}$
  $\mathbf{c}_y \leftarrow \mathbf{0} \in \mathbb{R}^{d_{(k)}} \ \forall \ y \in \mathcal{Y}$
  *# Phase A: First Session Adaptation with PETL*
  **for** every sample $(\boldsymbol{x}, y) \in \mathcal{D}_1$ **do**
    Collect $\phi_L(\boldsymbol{x})$
    Update PETL parameters
  *# Phase B: Continual Learning with LayUP*
  **for** task $t = 1, \dots, T$ **do**
    **for** every sample $(\boldsymbol{x}, y) \in \mathcal{D}_t$ **do**
      Collect $\Phi_{-k:}(\boldsymbol{x})$ (Eq. (5))
      $\mathbf{G} \leftarrow \mathbf{G} + \Phi_{-k:}(\boldsymbol{x}) \otimes \Phi_{-k:}(\boldsymbol{x})$
      $\mathbf{c}_y \leftarrow \mathbf{c}_y + \Phi_{-k:}(\boldsymbol{x})$
    *# Cf. Appendix A.1*
    Optimize $\lambda$ to compute $(\mathbf{G} + \lambda \boldsymbol{I})^{-1}$

---

## 4.3 Memory and runtime complexity comparisons

We compare the memory and runtime requirements of LayUP with the three competitive CL methods ADAM (Zhou et al., 2023b), SLCA (Zhang et al., 2023), and RanPAC (McDonnell et al., 2023) for a typical class count of $C = 200$ and AdaptFormer (Chen et al., 2022) as PETL method. We neglect the memory cost of the ViT backbone, the fully connected classification layer, and the class prototypes, as they are similar to all

baselines. LayUP stores $\sim$1M PETL parameters and an additional multi-layer Gram matrix in memory. The size of this layer depends on the choice of $k$ and has $d^2_{(k)} = (d_{L-k+1} + \cdots + d_{L-1} + d_L)^2$ entries (note that for the ViT-B/16 (Dosovitskiy et al., 2021) architecture, every layer outputs tokens of the same dimensionality, such that $d_l = 768 \; \forall \; l \in \{1, \ldots, L\}$). For $k = 6$, which we use in our final experiments, the matrix stores $\sim$21M values. RanPAC with random projection dimension $M = 10\,000$, as used throughout the original paper, updates a Gram matrix of size 100M and requires an additional $\sim$11M parameters for the random projection layer and PETL parameters. SLCA stores $C$ additional covariance matrices of size $d^2_L$ with a total of $\sim$118M entries. Finally, ADAM stores a second copy of the feature extractor, necessitating 84M additional parameters. Assuming a comparable memory cost for model parameters and matrix entries, our method reduces additional memory requirements by 81%, 82%, and 75% compared with RanPAC, SLCA, and ADAM, respectively.

Considering runtime complexity during training, ADAM makes updates to all parameters of a ViT-B/16 model during first task adaptation. SLCA performs full body adaptation of a ViT-B/16 backbone during slow learning and additionally performs Cholesky decomposition (Cholesky, 1924) on the class-wise covariance matrices for pseudo-rehearsal, which requires $C \cdot d^3_L \approx 10^{11}$ operations. RanPAC and LayUP only make updates to PETL parameters during training and perform matrix inversion during inference, albeit with differently constructed and sized Gram matrices. For $k = 6$, the matrix inversion needs $(d_{L-k+1} + \cdots + d_{L-1} + d_L)^3 \approx 10^{11}$ operations for LayUP and $10\,000^3 = 10^{12}$ for RanPAC, thus LayUP reduces RanPAC's runtime complexity during inference by up to 90%.

## 5 Experiments

In what follows, we conduct a series of experiments to assess our approach under various CL settings. We start with an overview of the datasets and implementation details in Sec. 5.1, followed by empirical comparisons of our method against recent strong baselines under CIL, DIL, and OCL settings in Sec. 5.2 and Sec. 5.3. Ablation studies are presented in Sec. 5.4. We explore the impact of different configurations of the last $k$ layers on classifier performance in Sec. 5.5 and investigate the range of classes that benefit from hidden features in Sec. 5.6. We conclude our empirical analysis by evaluating LayUP as a plug-in for other class-prototype methods in Sec. 5.7.

### 5.1 Datasets and implementation details

Following prior works (Wang et al., 2022b;c; Zhou et al., 2023b; Zhang et al., 2023; McDonnell et al., 2023), we experiment with two ViT-B/16 (Dosovitskiy et al., 2021) models, the former (**ViT-B/16-IN21K**) with self-supervised pre-training on the ImageNet-21K dataset (Ridnik et al., 2021), the latter (**ViT-B/16-IN1K**) with additional supervised fine-tuning on the ImageNet-1K dataset (Krizhevsky et al., 2012). During adaptation in the CIL and DIL settings, PETL parameters are trained using a batch size of 48 for 20 epochs, Adam (Kingma & Ba, 2015) with momentum for optimization, and learning rate scheduling via cosine annealing, starting with 0.03. We train five prompt tokens for VPT and use a bottleneck dimension of 16 for AdaptFormer. All baselines are rehearsal-free and trained on the same aforementioned ViT backbones.

For the CIL and OCL settings, the seven representative split datasets are CIFAR-100 (**CIFAR**) (Krizhevsky, 2009), ImageNet-R (**IN-R**) (Hendrycks et al., 2021a), ImageNet-A (**IN-A**) (Hendrycks et al., 2021b), CUB-200 (**CUB**) (Wah et al., 2011), OmniBenchmark (**OB**) (Zhang et al., 2022), Visual Task Adaptation Benchmark (**VTAB**) (Zhai et al., 2020), and Stanford Cars-196 (**Cars**) (Krause et al., 2013). We use $T = 10$ for all datasets, except VTAB, which has a commonly used task count of $T = 5$. Additional results for task counts $T = 5$ and $T = 20$ can be found in Appendix C.4. For the DIL setting, baselines are trained on Continual Deepfake Detection Benchmark Hard (**CDDB-H**) (Li et al., 2023) with $T = 5$, a sub-sampled version of DomainNet (**S-DomainNet**) (Peng et al., 2019) with $T = 6$, and the domain-incremental formulation of ImageNet-R (**IN-R (D)**) with $T = 15$. Detailed descriptions and summary statistics of all datasets can be found in Appendix B.

For comparison, we use the average accuracy (Lopez-Paz & Ranzato, 2017) metric $A_t = \frac{1}{t} \sum_{i=1}^{t} R_{t,i}$, with $R_{t,i}$ denoting the classification accuracy on the $i^{\text{th}}$ task after training on the $t^{\text{th}}$ task. Based on the

| Method | CIFAR | IN-R | IN-A | CUB | OB | VTAB | Cars |
|---|---|---|---|---|---|---|---|
| Joint FT (full) | 93.6 | 86.6 | 71.0 | 91.1 | 80.3 | 92.5 | 83.7 |
| Joint FT (linear probe) | 87.9 | 71.2 | 56.4 | 89.1 | 78.8 | 90.4 | 66.4 |
| L2P (Wang et al., 2022c) | 84.6 | 72.5 | 42.5 | 65.2 | 64.7 | 77.1 | 38.2 |
| DualPrompt (Wang et al., 2022b) | 81.3 | 71.0 | 45.4 | 68.5 | 65.5 | 81.2 | 40.1 |
| CODA-P (Smith et al., 2023a) | 86.3 | 75.5 | 74.5 | 79.5 | 68.7 | 87.4 | 43.2 |
| NMC+FSA (Janson et al., 2022) | 87.8 | 70.1 | 49.7 | 85.4 | 73.4 | 88.2 | 40.5 |
| ADAM (Zhou et al., 2023b) | 87.6 | 72.3 | 52.6 | 87.1 | 74.3 | 84.3 | 41.4 |
| SLCA (Zhang et al., 2023) | 91.5 | 77.0 | 59.8* | 84.7 | 73.1* | 89.2* | 67.7 |
| RanPAC (McDonnell et al., 2023) | **92.2** | 78.1 | 61.8 | **90.3** | **79.9** | 92.6 | 77.7 |
| **LayUP** | 92.0 | **81.4** | **62.2** | 88.1 | 77.6 | **93.3** | **82.5** |
| **Ablations** | | | | | | | |
| $k = 1$ | 90.8 | 78.8 | 60.4 | 86.7 | 72.0 | 92.4 | 74.9 |
| w/o FSA | 88.7 | 73.1 | 57.7 | 86.2 | 77.0 | 92.6 | 78.6 |
| $k = 1$, w/o FSA | 86.5 | 69.4 | 55.4 | 85.4 | 70.7 | 91.6 | 69.1 |
| LayNMC | 88.1 | 59.3 | 50.0 | 82.5 | 68.9 | 86.5 | 40.0 |
| NMC | 83.4 | 61.2 | 49.3 | 85.1 | 73.1 | 88.4 | 37.7 |

Table 1: **Comparison of prompting, backbone fine-tuning, and class-prototype methods for the CIL setting.** Results are taken from McDonnell et al. (2023) except results for SLCA marked with (*), which are reproduced using the officially released code. Best results per dataset are highlighted in bold.

| Method | CDDB-H | S-DomainNet | IN-R (D) |
|---|---|---|---|
| Joint FT (full) | 92.9 | 62.8 | 86.6 |
| Joint FT (linear probe) | 74.0 | 57.2 | 71.2 |
| NMC+FSA (Janson et al., 2022) | 49.8 | 44.0 | 76.3 |
| ADAM (Zhou et al., 2023b) | 70.7 | – | – |
| RanPAC (McDonnell et al., 2023) | 86.2 | 58.8 | 77.2 |
| **LayUP** | **88.1** | **59.4** | **80.0** |
| **Ablations** | | | |
| $k = 1$ | 84.0 | 54.2 | 77.8 |
| w/o FSA | 86.7 | 58.3 | 73.3 |
| $k = 1$, w/o FSA | 77.7 | 53.7 | 69.0 |
| LayNMC | 61.4 | 34.4 | 59.1 |
| NMC | 57.6 | 43.5 | 64.1 |

Table 2: **Comparison of class-prototype methods for the DIL setting.** Results for ADAM and RanPAC on CDDB-H are taken from McDonnell et al. (2023).

observations detailed in Sec. 4.1, utilizing features from the second half of the network layers for prototype construction emerges as an effective yet cost-efficient strategy in terms of memory and computational resources. Consequently, all experiments are conducted with $k = 6$, unless specified otherwise. For a comprehensive comparison of performance across different choices of $k$ for all datasets, refer to Sec. 5.5. In all experiments in the main paper, the reported metric is average accuracy $A_T$ after learning the last task for random seed 1993 and the best combination of PETL method and ViT backbone (similar to McDonnell et al. (2023); Zhou et al. (2023b)). We refer to Appendix C.1 for analysis of each $A_t$, forgetting measures, and variability across random initialization, and to Appendix C.3 for average accuracy over training for different PETL methods and pre-trained backbones in the CIL setting.

| Method | CIFAR | IN-R | IN-A | CUB | OB | VTAB | Cars |
|---|---|---|---|---|---|---|---|
| Sequential FT | 12.8 | 5.3 | 6.9 | 5.1 | 3.3 | 7.3 | 10.9 |
| NMC (Janson et al., 2022) | 83.4 | 61.2 | 49.3 | 85.1 | 73.1 | 88.4 | 37.7 |
| RanPAC$_{\lambda=0}$ (McDonnell et al., 2023) | **88.7** | 70.6 | 1.4 | 0.9 | 76.9 | 9.3 | 0.6 |
| RanPAC$_{\lambda=1}$ (McDonnell et al., 2023) | **88.7** | 70.6 | 0.9 | 0.7 | 76.9 | 9.0 | 0.6 |
| **LayUP**$_{\lambda=0}$ | **88.7** | **73.4** | 40.9 | 86.4 | **77.0** | 10.4 | 66.3 |
| **LayUP**$_{\lambda=1}$ | **88.7** | **73.4** | 51.6 | **86.9** | **77.0** | **92.9** | **77.5** |
| Ablations | | | | | | | |
| $k=1, \lambda=0$ | 86.5 | 69.6 | **55.6** | 85.4 | 72.8 | 92.2 | 69.2 |
| $k=1, \lambda=1$ | 86.5 | 69.6 | **55.6** | 85.4 | 76.3 | 92.3 | 69.2 |

Table 3: **Comparison of class-prototype methods for the OCL setting.** Results for RanPAC and NMC are reproduced according to the officially released code in McDonnell et al. (2023).

## 5.2 Performance in the CIL and DIL settings

In the CIL setting, we compare LayUP with several prompt learning, fine-tuning, and class-prototype methods for CL. We additionally report results for joint fine-tuning (FT) of the full backbone and a linear probe. As shown in Tab. 1, LayUP surpasses all baselines for four of the seven split datasets.

The four datasets—IN-R, IN-A, VTAB, and Cars—on which LayUP consistently outperforms all baseline models, exhibit two distinct characteristics: First, they possess a high domain gap relative to the pre-training ImageNet domain, as detailed in Sec. 5.5. Second, with the exception of IN-R, they contain significantly less training data compared to CIFAR, OB, and CUB (cf. Appendix B). The former confirms our initial hypothesis that intermediate representations are more domain-invariant and consequently more robust to large distributional shifts from the source to the target domain. The latter indicates that especially in the low-data regime, in contrast to other methods that tend to overfit to the target domain, LayUP constructs class prototypes and decision boundaries that generalize well even if the amount of training data is scarce compared with the data used for ViT pre-training. It is further noteworthy that RanPAC, which is the only baseline that LayUP does not consistently outperform, is considerably more expensive regarding both memory and computation (cf. Sec. 4.3).

Results for the DIL setting are reported in Tab. 2. We compare our method with several strong class-prototype methods for CL and, similar to the CIL comparison, we additionally report results for joint fine-tuning of the backbone and a linear probe. LayUP consistently outperforms all baselines. These results confirm that our method effectively utilizes the domain-invariant information contained in the low- and mid-level intermediate features of the pre-trained model, thereby enhancing its robustness to domain shifts.

## 5.3 Performance in the OCL setting

We are interested in assessing the performance of our method in the challenging OCL setting, where only a single pass over the continual data stream is allowed. All baselines are class-prototype methods for continual learning on a frozen embedding network (thus we omit FSA stages for NMC, RanPAC, and our approach), except for sequential fine-tuning, where we update the parameters of the pre-trained ViT via cross-entropy loss and Adam optimizer during a single epoch. Note that NMC exhibits identical behavior in both CIL and OCL settings, as it uses each training sample only once for running-mean calculation, consequently producing the same outcomes as the ablation baseline in Tab. 1. Given that the choice of the ridge regression parameter $\lambda$, as utilized in RanPAC and our method, necessitates prior knowledge about the downstream continual data—a requirement unmet in realistic streaming learning settings—we opt for comparison with two simplified versions of Gram matrix inversion $(\boldsymbol{G} + \lambda \boldsymbol{I})^{-1}$ (cf. Eq. (3) and Eq. (6)): In the first variant, we omit $\lambda$-based regularization completely, such that $\lambda = 0$ and the Gram matrix is inverted without regularization (i.e., $\boldsymbol{G}^{-1}$). In the second variant, e.g., as used in Panos et al. (2023), we choose $\lambda = 1$, such that we linearly transform identity-regularized class prototypes (i.e., $(\boldsymbol{G} + \boldsymbol{I})^{-1}$).

| $k$ | CIFAR | IN-R | IN-A | CUB | OB | VTAB | Cars |
|---|---|---|---|---|---|---|---|
| 1 | 89.2 | 79.2 | 60.8 | 85.3 | 71.6 | 92.7 | 75.5 |
| 2 | 90.1 | 79.9 | 62.2 | 86.9 | 73.9 | 90.7 | 78.7 |
| 3 | 90.2 | 80.7 | 62.5 | 87.3 | 74.5 | 93.2 | 81.4 |
| 4 | 90.4 | 81.1 | 62.0 | 87.6 | 76.7 | 93.1 | 81.7 |
| 5 | 90.7 | 81.5 | 62.8 | 87.3 | 77.0 | 93.0 | 82.4 |
| 6 | 91.0 | 81.2 | 62.2 | 87.3 | 77.5 | 92.2 | 82.5 |
| 7 | 90.9 | **81.6** | **63.4** | 87.5 | 77.7 | 93.4 | **82.6** |
| 8 | 90.8 | 81.4 | 62.8 | 87.7 | 77.2 | 92.1 | 82.3 |
| 9 | 91.0 | 81.5 | 62.1 | 87.4 | **78.3** | 92.0 | **82.6** |
| 10 | 90.9 | 81.4 | **63.4** | 87.9 | 77.4 | **94.0** | 81.5 |
| 11 | **91.1** | 81.5 | 62.0 | 87.6 | 78.1 | 93.7 | 81.4 |
| 12 | 90.8 | 80.8 | 62.8 | **88.0** | 77.8 | 93.9 | 82.3 |

Table 4: **Comparison of maximum representation depths.** LayUP performance for different values of $k$ for a pre-trained ViT-B/16-IN1K and FSA with AdaptFormer. The 1st , 2nd , and 3rd highest scores and the 1st , 2nd , and 3rd lowest scores are highlighted.

As indicated in Tab. 3, unrestricted fine-tuning of the backbone is detrimental to the generalizability of the pre-trained embeddings and leads to forgetting and low performance. Consequently, the sequential FT baseline is outperformed by class-prototype methods in the OCL setting for most benchmarks. With the exception of VTAB, LayUP is largely robust to missing regularization ($\lambda = 0$) and maintains a high performance across all benchmarks for regularization with $\lambda = 1$. On the contrary, RanPAC exhibits high variability in performance across benchmarks, occasionally resulting in near-zero accuracy. This outcome is likely due to the high-dimensional prototypes and Gram matrix in RanPAC, obtained after random projections, overfitting the training data, which consequently hampers their ability to generalize. LayUP's superior robustness in the online learning setting renders it a more favorable approach for continual learning scenarios where the length of the input stream and the nature of the data might not be known in advance.

## 5.4 Ablation study

The purpose of the following ablation study is to demonstrate the advantages of leveraging multi-layer representations over solely utilizing features from the final representation layer (i.e., $k = 1$) for class-prototype construction in CL. As LayUP integrates multi-layer representations with second-order statistics via Gram matrix inversion as well as first session adaptation in the CIL and DIL settings, we consider three configurations for ablating multi-layer representations: (i) in the standard setting with both FSA and second-order statistics (LayUP vs. $k = 1$), (ii) excluding the FSA stage, i.e., omitting phase A in Alg. 1 (w/o FSA vs. $k = 1$, w/o FSA), and (iii) excluding FSA and second-order statistics, which corresponds to traditional nearest-mean classifiers (LayNMC vs. NMC). Results for the baselines constructed for settings (i)-(iii) can be found in the bottom sections of Tab. 1 and Tab. 2. For (i), enriching the final layer with intermediate features consistently results in absolute performance gains ranging from 1.2% to 5.7% (CIL) and 2.2% to 5.2% (DIL). For (ii), except for the VTAB dataset in the CIL setting, leveraging multi-layer features yields absolute accuracy improvements of up to 9.5% (CIL) and 9.0% (DIL). Lastly, for (iii), LayNMC does not show superior performance to the NMC baseline. Such results indicate that the benefits of intermediate features diminish when not decorrelated via Gram matrix inversion, as they contain more lower-level noisy information than final layer representations.

In the lower section of Tab. 3, further comparisons are made between ablated versions of LayUP and prototypes using final representations ($k = 1$) in the OCL setting for two selected values of the regularization parameter $\lambda$. LayUP generally outperforms its ablated versions, although it demonstrates decreased performance in low-data benchmarks such as IN-A and VTAB, particularly when omitting regularization ($\lambda = 0$). This decline in performance may stem from the increased dimensionality of class prototypes and the Gram matrix due to the addition of intermediate representations, thereby elevating the risk of overfitting. Consequently, the

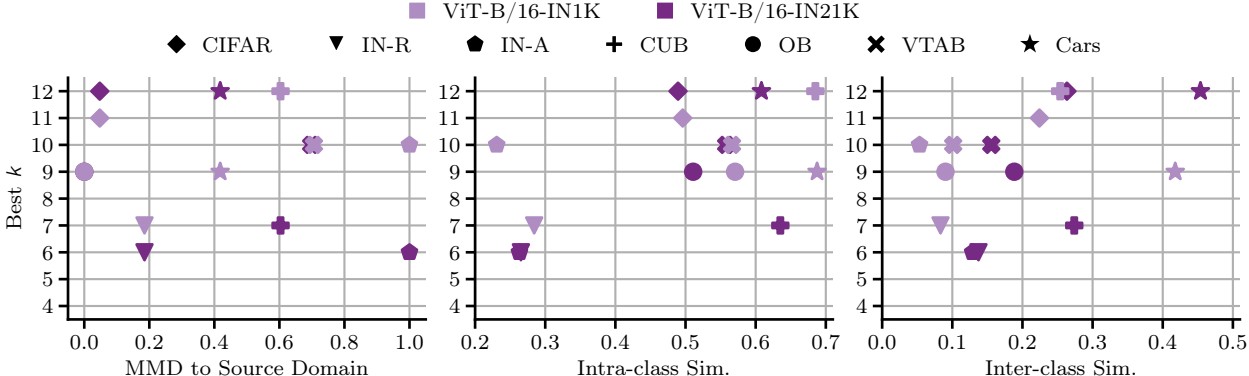

Figure 4: **Choice of k for different dataset characteristics.** Multi-layer representation depth $k$ that yields highest accuracy vs. normalized MMD between each dataset and the ImageNet pre-training domain, represented by *mini*ImageNet dataset (*left*), intra-class similarity (*center*), and inter-class similarity (*right*).

absence of regularization, combined with limited training data, may render multi-layer features detrimental. However, in any other case, they enhance classification accuracy by 0.6% to 8.3%.

## 5.5 Analysis of multi-layer representation depth

To provide more insights into the impact of the multi-layer representation depth on LayUP performance, we plot in Tab. 4 the average accuracy after training for different choices of $k$ using a ViT-B/16-IN1K backbone and AdaptFormer as PETL method. The final-layer classifier ($k = 1$) exhibits the lowest performance in comparison to all other $k$ values across all datasets, except for the split VTAB dataset. Conversely, LayUP configurations with medium or large $k$ values almost consistently outperform those with small $k$ values, demonstrating the broad advantages of leveraging representations from multiple intermediate layers for class-prototype construction. While the optimal choice of $k$ varies across datasets, the performance gains generally diminish with increasing $k$ values. For instance, the average accuracy gain from choosing $k = 6$ over $k = 1$ is 2.7%, whereas it is only 0.4% when choosing $k = 12$ over $k = 6$. This confirms the findings made in Sec. 4.1, suggesting that a value of $k = 6$ might generally suffice to achieve a significant performance gain over last-layer class-prototype methods while requiring significantly less computational and memory resources compared with $k = 12$.

We further aim to determine whether prior knowledge about the characteristics of downstream CL data can inform the optimal choice of maximum multi-layer representation depth with respect to overall performance. Specifically, we identify the value of $k$ that maximizes accuracy on the test set after running Alg. 1 with AdaptFormer as PETL method for each split dataset used in Sec. 5.2 and for each pre-trained ViT model. We plot the highest performing $k$ per dataset against three measures: (i) the degree of domain gap to the pre-training ImageNet domain, measured by Maximum Mean Discrepancy (MMD); (ii) the intra-class similarity, measured by the average pairwise cosine similarity between training image feature vectors of the same class; and (iii) the inter-class similarity, measured by the average pairwise cosine similarity between image feature vectors of disjoint classes. For (i), we follow Panos et al. (2023) and use the *mini*ImageNet dataset (60K images) as a proxy for the ImageNet-1K (1.3M images) and ImageNet-21K (14M images) pre-training datasets to reduce computational overhead when calculating pairwise distances. A large MMD value indicates that two datasets have significantly different statistical properties, thus signifying a large domain gap between the source and target domains.

As shown in Fig. 4, there is no clear relationship between the MMD and the optimal choice of $k$. This suggests that the degree of domain gap from the source domain of a pre-trained model is not a reliable indicator for determining the multi-layer representation depth that maximizes performance. However, a positive trend is observed between the optimal $k$ and both intra-class and inter-class similarity.

First, datasets with high intra-class and inter-class similarity (e.g., Cars and CUB) benefit from a high value of $k$. These datasets typically focus on fine-grained classification of natural images within a specific domain. Second, datasets with medium intra-class similarity and low inter-class similarity (e.g., VTAB, OB, or CIFAR) benefit from a medium-to-high value of $k$. These datasets are usually composed of multiple specialized natural-image datasets, which can belong to very distinct domains, thus explaining the low inter-class similarity. Finally, datasets with low intra-class and inter-class similarity (e.g., IN-A or IN-R) benefit from a medium value of $k$. These datasets often consist of atypical or stylized image examples that differ from the natural images in ImageNet and are thus prone to be distributed unsystematically in the feature space.

Although the maximum performance for most configurations in Fig. 4 is achieved by concatenating features from the majority of layers in the pre-trained model, the performance differences between various choices of $k$ can be minimal in some cases (as shown in Tab. 4) and may fluctuate with different random initializations. Furthermore, as previously discussed, performance gains tend to saturate as $k$ increases. Concurrently, larger $k$ values are associated with higher memory usage and increased runtime complexity during inference. Therefore, selecting an appropriate $k$ requires careful consideration of the specific demands and constraints across different dimensions.

### 5.6 How *universal* are multi-layer prototypes?

We aim to assess the universality of multi-layer prototypes for classification, specifically, the breadth of classes benefiting from LayUP. To this end, we report the proportion of classes for each dataset whose test accuracy scores are higher, equal, or lower with LayUP compared to the final-layer ridge classifier (cf. Eq. (3)). Beyond merely counting the classes that are better predicted by either method, we also examine the extent of the benefit one method has over the other per improved class. Consequently, we calculate the average difference in accuracy scores between LayUP and the final-layer classifier ($k = 1$), and vice versa, for each class that shows improvement.

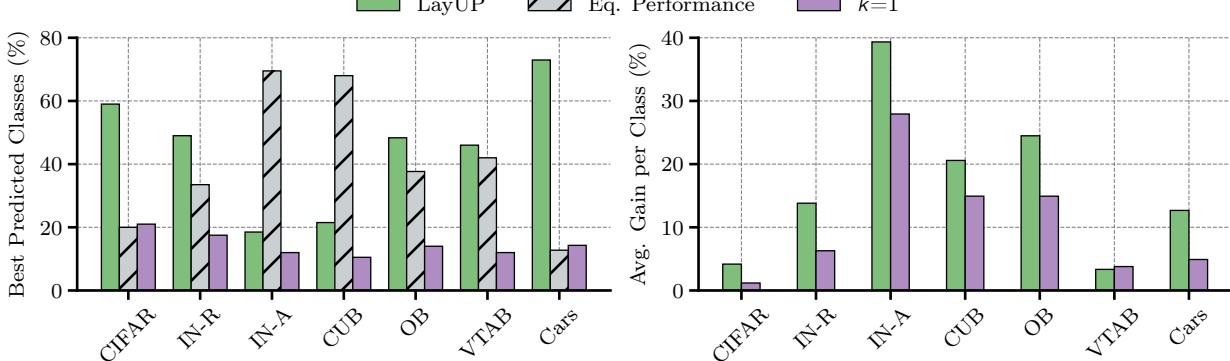

Figure 5: **LayUP vs. classification from the final representation layer.** *Left:* Percentage of classes per dataset that are better classified with LayUP (with $k = 6$) compared with the last layer ridge classifier (equivalent to LayUP with $k = 1$). *Right:* Difference of absolute accuracy per improved class between LayUP and the final-layer classifier ($k = 1$), and vice versa.

Fig. 5 illustrates that a higher percentage of classes (between 18% and 72%) is more effectively classified upon introducing intermediate representations to class-prototype construction. Although the difference in the number of improved classes is not as pronounced for IN-A and CUB compared with all other benchmarks, the relative difference in accuracy per improved class between LayUP and the final representation layer classifier reaches as high as 39% for IN-A and 22% for CUB. This indicates that for these two datasets, the introduction of intermediate representations significantly benefits a few classes rather than slightly benefiting a large number of classes, as observed with the other benchmarks. Despite the variability with respect to the number of classes per dataset that benefit from LayUP and the extent of the benefit, our results demonstrate the universal applicability of multi-layer prototypes across a broad spectrum of tasks, classes, or domains.

| Method | CIFAR | IN-R | IN-A | CUB | OB | VTAB | Cars |
|---|---|---|---|---|---|---|---|
| ADAM | 87.6 | 72.3 | 50.5 | 87.1 | 74.3 | 84.3 | 51.3 |
| ADAM w/ LayUP ($k = 4$) | 91.0 | 81.8 | 63.3 | 87.5 | 78.1 | 93.9 | **82.4** |
| ADAM w/ LayUP ($k = 6$) | **91.4** | 81.7 | **63.8** | **87.8** | **79.0** | **94.7** | 82.2 |
| ADAM w/ LayUP ($k = 12$) | 91.3 | **82.0** | **63.8** | 87.5 | 78.1 | 94.6 | 81.7 |

(a) LayUP + ADAM (Zhou et al., 2023b)

| Method | CIFAR | IN-R | IN-A | CUB | OB | VTAB | Cars |
|---|---|---|---|---|---|---|---|
| RanPAC | 90.7 | 78.0 | 58.2 | 88.5 | 76.9 | 92.6 | 67.5 |
| RanPAC w/ LayUP ($k = 4$) | **91.1** | 82.4 | 61.5 | 88.7 | **79.0** | 94.0 | 81.9 |
| RanPAC w/ LayUP ($k = 6$) | **91.1** | **82.8** | 63.4 | 88.6 | 78.3 | 93.4 | **82.3** |
| RanPAC w/ LayUP ($k = 12$) | **91.1** | 82.4 | **64.1** | **88.9** | 78.3 | **94.1** | 81.7 |

(b) LayUP + RanPAC (McDonnell et al., 2023)

Table 5: **Combination of LayUP with other class-prototype methods.** The reported average accuracy scores (%) pertain to the CIL setting as detailed in Sec. 5.1. All baselines were trained using ViT-B/16-IN1K as the pre-trained model and AdaptFormer as the PETL method. Results for RanPAC and ADAM, which were not reported for this specific configuration, have been reproduced using their officially released code repositories. Note that the results for RanPAC and ADAM may differ slightly from those presented in Tab. 1, as the latter reports the best among all configurations of PETL methods and ViT models.

## 5.7  Combination with other class-prototype methods

We aim to investigate the effectiveness of using intermediate representations, as employed in LayUP, as a plug-in to enhance other prototype-based methods for CL. Specifically, we incorporate multi-layer representations into two class-prototype methods: ADAM and RanPAC. Details on these methods can be found in Zhou et al. (2023b) and McDonnell et al. (2023). For LayUP combined with ADAM, we aggregate concatenated features from the last $k$ layers of both the pre-trained and first session adapted ViT to construct prototypes. During inference, instead of using cosine similarity matching as done in ADAM (cf. Eq. (2)), we apply Gram matrix inversion following the approach in LayUP (cf. Eq. (6)). This decision is informed by the results from Sec. 5.4, which suggest that multi-layer representations are insufficiently effective for prototype matching when relying solely on first-order feature statistics. For LayUP combined with RanPAC, concatenated features from the last $k$ layers of the first session adapted ViT are fed to the random projection layer, which has an output dimensionality of $M = 10\,000$.

The results for the combinations with ADAM and RanPAC are shown in Tab. 5a and Tab. 5b, respectively. Integrating LayUP with other class-prototype methods consistently improves performance across all datasets. However, the magnitude of these improvements varies, with the CUB and CIFAR datasets showing the least pronounced enhancements ($\uparrow 0.4\%$ for LayUP integrated with RanPAC) and the Cars dataset exhibiting the most substantial gains ($\uparrow 31.1\%$ for LayUP integrated with ADAM). Although the optimal value of $k$ for maximizing performance differs across datasets, the overall performance does not significantly vary with different $k$ values. This suggests that a small, resource-efficient choice of $k$ is sufficient to enhance existing class-prototype methods when integrated with LayUP.

## 6  Conclusion

In this paper, we propose LayUP, a simple yet effective rehearsal-free class-prototype method for continual learning with pre-trained models. LayUP leverages multi-layer representations of a pre-trained feature extractor to increase robustness and generalizability, especially under large domain gaps and in low-data regimes. It further computes second-order feature statistics to decorrelate class prototypes, combined with parameter-efficient first session adaptation. Extensive experiments across a range of image classification datasets and continual learning settings demonstrate that LayUP performs strongly against competitive

baselines while requiring significantly less memory and computational complexity. Beyond that, our method can be used as a versatile plug-in to improve existing class-prototype methods. In future work, we will further investigate integrating class-prototype methods with continual adaptation of pre-trained models beyond first session adaptation, which is particularly beneficial for learning under multiple distributional shifts. Multi-layer representations serve as a potent source of knowledge that is inherently available in pre-trained models. Based on the findings made in this work, we encourage further exploration into making explicit use of these representations (i.e., *reading between the layers*) in future approaches to continual learning.

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

## Appendix

## A  Training and Implementation Details

### A.1  Optimization of ridge regression parameter $\lambda$

The optimization process for selecting the regression parameter $\lambda$ (cf. Alg. 1) proceeds as follows: For each task $t$, we stratify the training data by ground truth labels and perform a four-fold split. For each split, we temporarily update $\boldsymbol{G}$ and $\boldsymbol{c}_y$ for all $y \in \mathcal{Y}_t$, and calculate the accuracy for values of $\lambda \in \{10^{-8}, 10^{-4}, 10^{-3.5}, 10^{-3}, 10^{-2.5}, 10^{-2}, 10^{-1.5}, 10^{-1}, 10^{-0.5}, 10^0, 10^{0.5}, 10^1, 10^{1.5}, 10^2, 10^{2.5}, 10^3\}$ to determine which yields the highest per-sample accuracy on the remaining training data fold. We then calculate the mean accuracy across all folds for each $\lambda$ and select the value that results in the highest overall accuracy. Finally, we update $\boldsymbol{G}$ and $\boldsymbol{c}_y$ for all $y \in \mathcal{Y}_t$ using the complete training dataset for task $t$, and repeat the process for task $t + 1$. Note that this procedure, when performed at time $t$, does not require access to data from any prior task, thus maintaining full compatibility with the rehearsal-free CL setting.

### A.2  Data augmentation

During CL training, data augmentation is applied to all datasets, incorporating random cropping of the images to varying sizes, ranging from 70% to 100% of their original dimensions, while maintaining an aspect ratio between 3:4 and 4:3. After resizing, images are randomly flipped horizontally, and brightness, contrast, saturation, and hue are varied randomly within a 10% range. Finally, the images are center-cropped to 224×224 pixels for all datasets, except for CIFAR-100, where images are directly resized from the original 32×32 to 224×224 pixels. During inference, images of all datasets are resized to 224x224 pixels without further modification.

### A.3  Compute resources

All experiments in this work were conducted on an Ubuntu system version 20.04.6 with a single NVIDIA GeForce RTX 3080 Ti (12GB memory) GPU.

## B  Datasets

We summarize the datasets compared in the main experiments in Tab. 6.

**Datasets for the CIL/OCL settings.**  CIFAR-100 (**CIFAR**) comprises 100 classes of natural images across various domains and topics, aligning relatively closely with the pre-training domains of ImageNet-1K and ImageNet-21K in terms of distribution. ImageNet-R (**IN-R**) consists of image categories that overlap with ImageNet-1K, yet it features out-of-distribution samples for the pre-training dataset, including challenging examples or newly collected data of various styles. Similarly, ImageNet-A (**IN-A**) shares categories with ImageNet-1K but encompasses real-world, adversarially filtered images designed to deceive existing classifiers pre-trained on ImageNet. The Caltech-UCSD Birds-200-2011 (**CUB**) dataset is a specialized collection of labeled images of 200 bird species, encompassing a diverse range of poses and backgrounds. OmniBenchmark (**OB**) serves as a compact benchmark designed to assess the generalization capabilities of pre-trained models across semantic super-concepts or realms, encompassing images of 300 categories that represent distinct concepts. The Visual Task Adaptation Benchmark (**VTAB**) as used in our work is a composition of the five datasets Resisc45 (Cheng et al., 2017), DTD (Cimpoi et al., 2014), Pets (Parkhi et al., 2012), EuroSAT (Helber et al., 2019), and Flowers (Nilsback & Zisserman, 2006) and is commonly split into $T = 5$ tasks in the CL version. The Stanford Cars-196 (**Cars**) dataset comprises images of cars belonging to one of 196 unique combinations of model and make.

**Datasets for the DIL setting.**  The Continual Deepfake Detection Benchmark Hard (**CDDB-H**) is a framework designed to evaluate the performance of deepfake detection models under continuously changing attack scenarios (*gaugan, biggan, wild, whichfaceisreal,* and *san*). DomainNet is a multi-source domain

| Setting | Name | Original | CL Formulation | $T$ | $N_{\text{train}}$ | $N_{\text{val}}$ | $C$ |
|---|---|---|---|---|---|---|---|
| | **CIFAR** | Krizhevsky (2009) | Rebuffi et al. (2017b) | 10 | 50 000 | 10 000 | 100 |
| | **IN-R** | Hendrycks et al. (2021a) | Wang et al. (2022b) | 10 | 24 000 | 6 000 | 200 |
| | **IN-A** | Hendrycks et al. (2021b) | Zhou et al. (2023b) | 10 | 6 056 | 1 419 | 200 |
| CIL/OCL | **CUB** | Wah et al. (2011) | Zhou et al. (2023b) | 10 | 9 465 | 2 323 | 200 |
| | **OB** | Zhang et al. (2022) | Zhou et al. (2023b) | 10 | 89 668 | 5 983 | 300 |
| | **VTAB** | Zhai et al. (2020) | Zhou et al. (2023b) | 5 | 1 796 | 8 619 | 50 |
| | **Cars** | Krause et al. (2013) | Zhang et al. (2023) | 10 | 8 144 | 8 041 | 196 |
| | **CDDB-H** | Li et al. (2023) | Li et al. (2023) | 5 | 16 068 | 10 706 | 2 |
| DIL | **S-DomainNet** | Peng et al. (2019) | Peng et al. (2019) | 6 | 20 624 | 176 743 | 345 |
| | **IN-R (D)** | Hendrycks et al. (2021a) | – | 15 | 24 000 | 6 000 | 200 |

Table 6: Overview of datasets. Original publication, split formulation for CL setting, number of tasks ($T$) in the main experiments in Sec. 5, number of training samples ($N_{\text{train}}$), number of validation samples ($N_{\text{val}}$), and number of classes ($C$). Although IN-R was originally proposed for domain generalization tasks, we are not aware of any prior work to use it in the DIL setting.

adaptation benchmark, comprising the six image style domains *real*, *quickdraw*, *painting*, *sketch*, *infograph*, and *clipart*. As the original training dataset of DomainNet comprises more than 400 000 samples, we use a sub-sampled version of DomainNet (**S-DomainNet**), that caps the number of training samples at 10 per class and domain, while maintaining the original validation set unchanged. Finally, with the domain-incremental split of ImageNet-R (**IN-R (D)**), a sequence of fifteen image style domains is learned (*sketch*, *art*, *cartoon*, *deviantart*, *embroidery*, *graffiti*, *graphic*, *misc*, *origami*, *painting*, *sculpture*, *sticker*, *tattoo*, *toy*, and *videogame*).

# C   Additional Results

We use the following two metrics in our work for experimental evaluation: *Average accuracy $A_t$* (following the definition of Lopez-Paz & Ranzato (2017)) and *average forgetting $F_t$* (following the definition of Chaudhry et al. (2018)). Average accuracy is defined as

$$A_t = \frac{1}{t} \sum_{i=1}^{t} R_{t,i}, \tag{8}$$

where $R_{t,i}$ denotes the classification accuracy on task $i$ after training on task $t$. Using the same notion of $R_{t,i}$, average forgetting is defined as

$$F_t = \frac{1}{t-1} \sum_{i=1}^{t-1} \underset{t' \in \{1,...,t-1\}}{\arg\max} R_{t',i} - R_{t,i} \tag{9}$$

As the number of different classes to choose from in a CIL and an OCL setting grows as new tasks are being introduced, $A_t$ tends to decrease and $F_t$ tends to rise over the course of training. While we only report on average accuracy metric in the main paper, the experimental results in Appendix C.1, Appendix C.2, and Appendix C.3 are reported both with respect to average accuracy and average forgetting.

## C.1   Variability across seeds and performance over time

As the choice of the random seed has an influence on both the task order and PETL parameter initialization, we calculate average accuracy and forgetting with standard error for all seven datasets over the course of class-incremental training (phase B in Alg. 1) using seeds 1993-1997. Results are shown in Fig. 6. There are almost no differences between final performance after the last task across seeds, which indicates LayUP being robust to different task orders and initialization of PETL parameters. The highest variability can be observed for the VTAB benchmark, where we have $T = 5$ and classes belong to very distinct domains. Therefore, the task order–and consequently the set of classes that the representations are adjusted to during first session

adaptation–has a greater influence on the generalization capabilities of the model during CL. As expected, the average forgetting increases as more tasks are being introduced, as the model has more classes to tell apart during inference.

## C.2   Experiments with different layer choice $k$

To analyze the learning behavior over time in the CIL setting among different choices of maximum representation depth $k$ for prototype construction, we plot average accuracy and forgetting for $k = 1$ (last layer only), $k = 6$, and $k = 12$ (all layers) in Fig. 7. Clearly, the classification performance based on last layer representations only is inferior to multi-layer representations as used in LayUP, as it yields both a lower accuracy and a higher forgetting rate. At the same time, there is no performance difference between $k = 6$ and $k = 12$ evident, indicating that the model's early layers do not contribute meaningful knowledge to the classification process, yet they do not detriment it either. Considering that the choice of $k$ is subject to a trade-off between performance gain and both memory and computational costs, the results corroborate $k = 6$, as used in the main experiments detailed in Sec. 5, to be a reasonable choice.

## C.3   Experiments with different backbones and PETL methods

Following prior works (Zhou et al., 2023b; McDonnell et al., 2023), we experiment with different PETL methods and ViT-B/16 models for LayUP, as the additional benefit from additional fine-tuning of the representations differs depending on the characteristics of the downstream domain (Panos et al., 2023). Results are shown in Fig. 8.

In all but one dataset, we found adapter methods (AdaptFormer) to be superior to be superior to prompt learning (VPT) or feature modulation (SSF). However, there is no combination of PETL method and pre-trained model that generally outperforms all other variants. Such results confirm the findings of (Panos et al., 2023) and underline the importance of considering different pre-training schemes and strategies for additional fine-tuning.

## C.4   Experiments with different task counts $T$

To show whether a benefit of multi-layer representations can be observed for different task counts, we compare average accuracy scores after training for six different datasets, three different task counts $T \in \{5, 10, 20\}$, and three different choices of $k$ last layers for class prototype generation. $k = 1$ corresponds to classification only based on the last layer (i.e., final) representations of the backbone, as it is done in prior work. $k = 6$ means to concatenate layer-wise features from the latter half of the network layers. Finally, $k = 12$ uses concatenated features of all layers of the pre-trained ViT for classification. Results are presented in Tab. 7.

Performance scores across task counts and datasets are consistently higher for multi-layer representations ($k = 6$ and $k = 12$) compared with only final representations ($k = 1$). However, there is no considerable difference in performance between $k = 6$ and $k = 12$. Thus, incorporating intermediate representations from layers in the second half of the model provides sufficient information for class separation, while features from early layers contribute minimally. This supports the observations noted in the preliminary analysis in Sec. 4.1.

| $k$ | $T$ | CIFAR | IN-R | IN-A | CUB | OB | Cars |
|---|---|---|---|---|---|---|---|
|  | 5 | 89.5 | 80.4 | 61.9 | 85.0 | 72.4 | 75.1 |
| 1 | 10 | 89.2 | 79.2 | 60.8 | 85.3 | 71.6 | 75.5 |
|  | 20 | 88.4 | 77.3 | 57.1 | 83.6 | 72.2 | 75.6 |
|  | 5 | 91.8 | 82.8 | 64.0 | 87.1 | 77.4 | 81.7 |
| 6 | 10 | 91.0 | 81.2 | 62.2 | 87.3 | 77.5 | 82.5 |
|  | 20 | 88.8 | 79.9 | 59.6 | 85.9 | 77.4 | 82.2 |
|  | 5 | 91.6 | 83.2 | 64.0 | 87.4 | 78.3 | 82.1 |
| 12 | 10 | 90.8 | 80.8 | 62.8 | 88.0 | 77.8 | 82.3 |
|  | 20 | 88.9 | 80.2 | 60.0 | 86.0 | 78.1 | 82.8 |

Table 7: Average accuracy (%) after training: Comparison of different task counts $T$ for $k = 1$ (prototype construction from last layer only), $k = 6$ (prototype construction from second half of the network layers, as utilized in Sec. 5), and $k = 12$ (prototype construction from all network layers). Scores listed are for AdaptFormer and ViT-B/16-IN1K. VTAB is omitted from the comparison due to its fixed number of datasets from which tasks are constructed ($T = 5$).

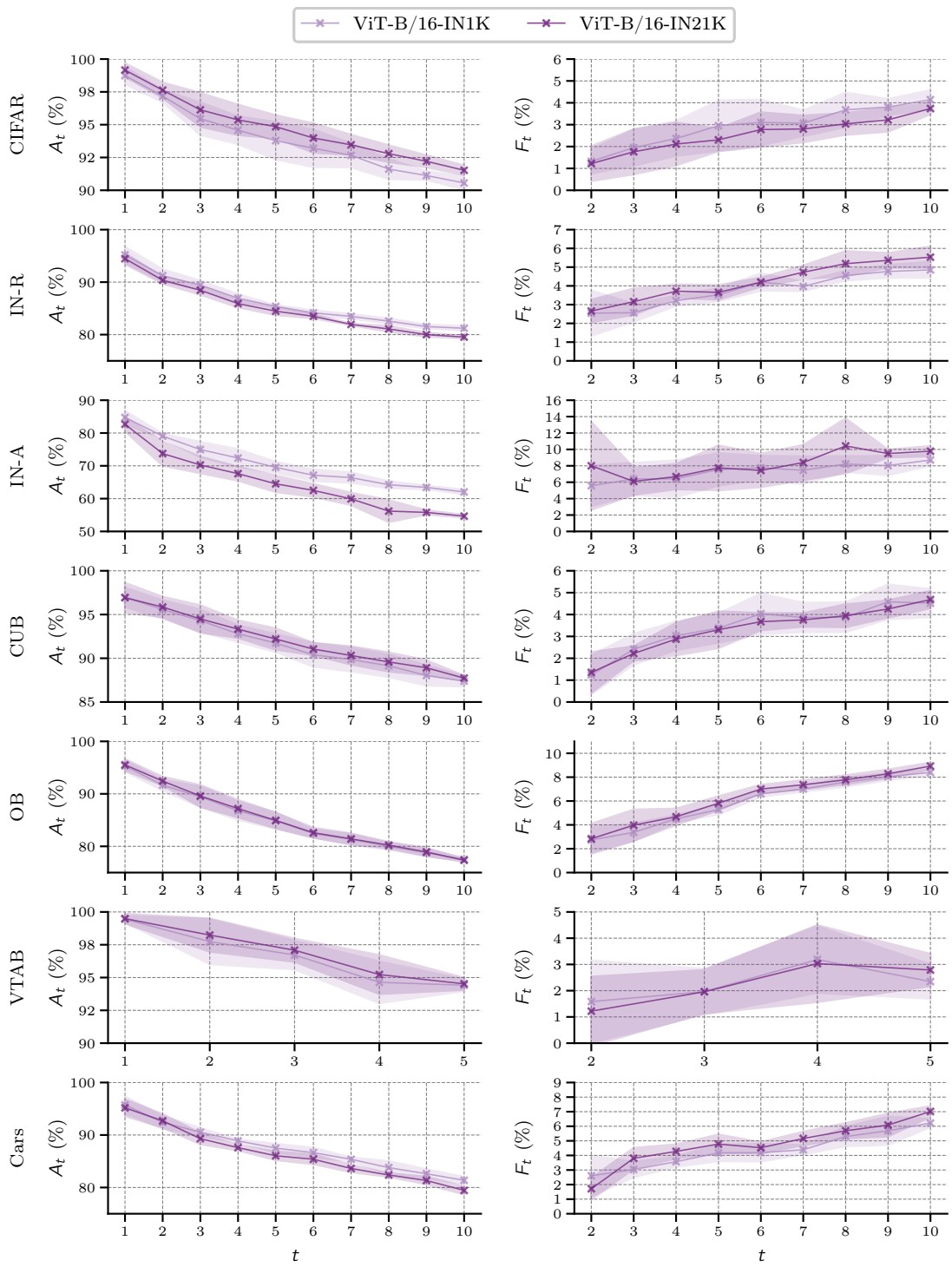

Figure 6: Average accuracy (*left*) and average forgetting (*right*) after training on each task $t$ in the CIL setting: Variability across random seeds for each ViT-B/16 models after first session training with AdaptFormer as PETL method. Results are reported for seeds 1993-1997 to ensure reproducibility with the resulting standard error indicated by shaded area.

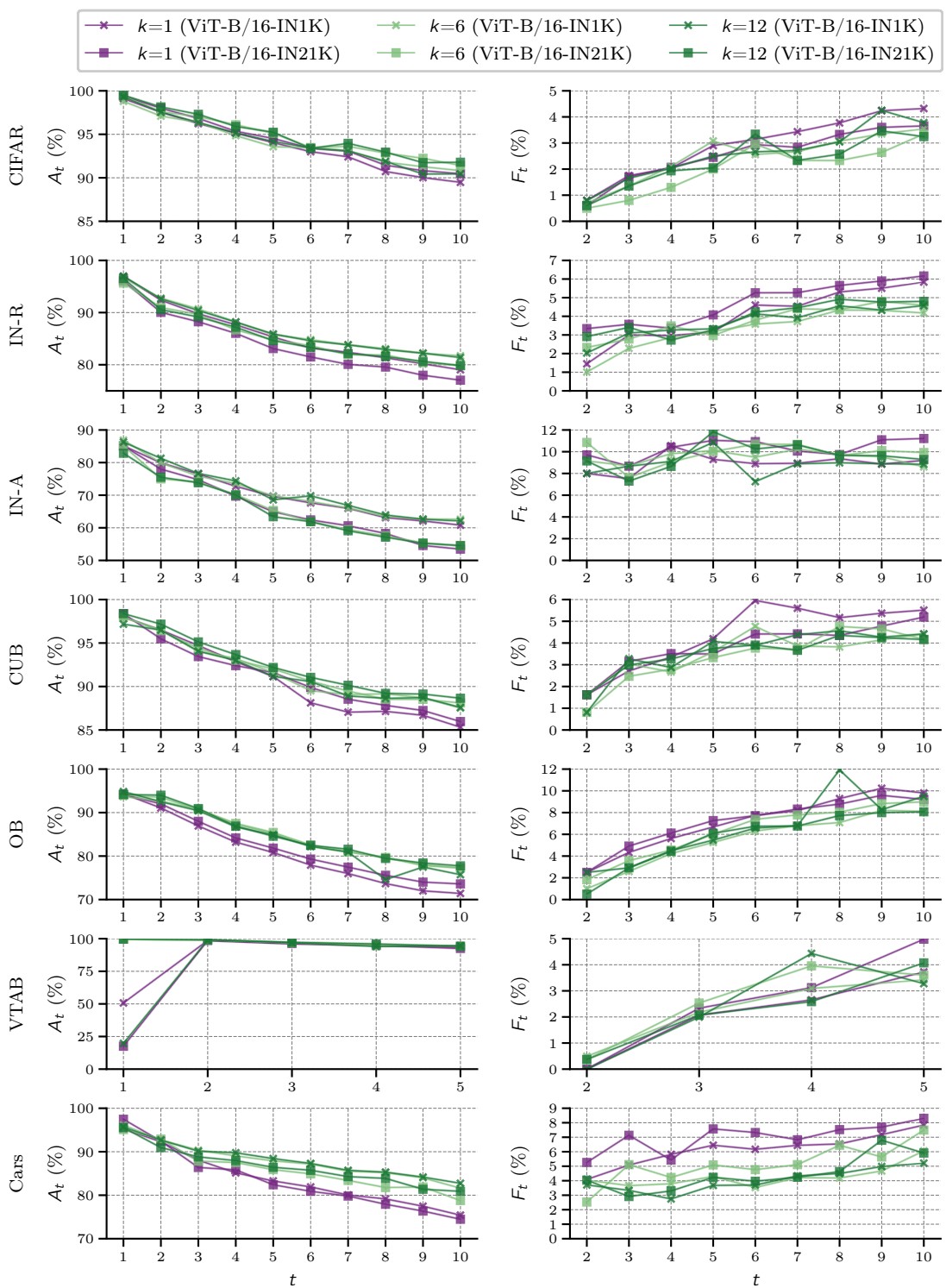

Figure 7: Average accuracy (*left*) and average forgetting (*right*) after training on each task $t$ in the CIL setting: Comparison of different choices of $k$ last layers for prototype construction ($k \in \{1, 6, 12\}$) and ViT-B/16 models after first session training with AdaptFormer as PETL method.

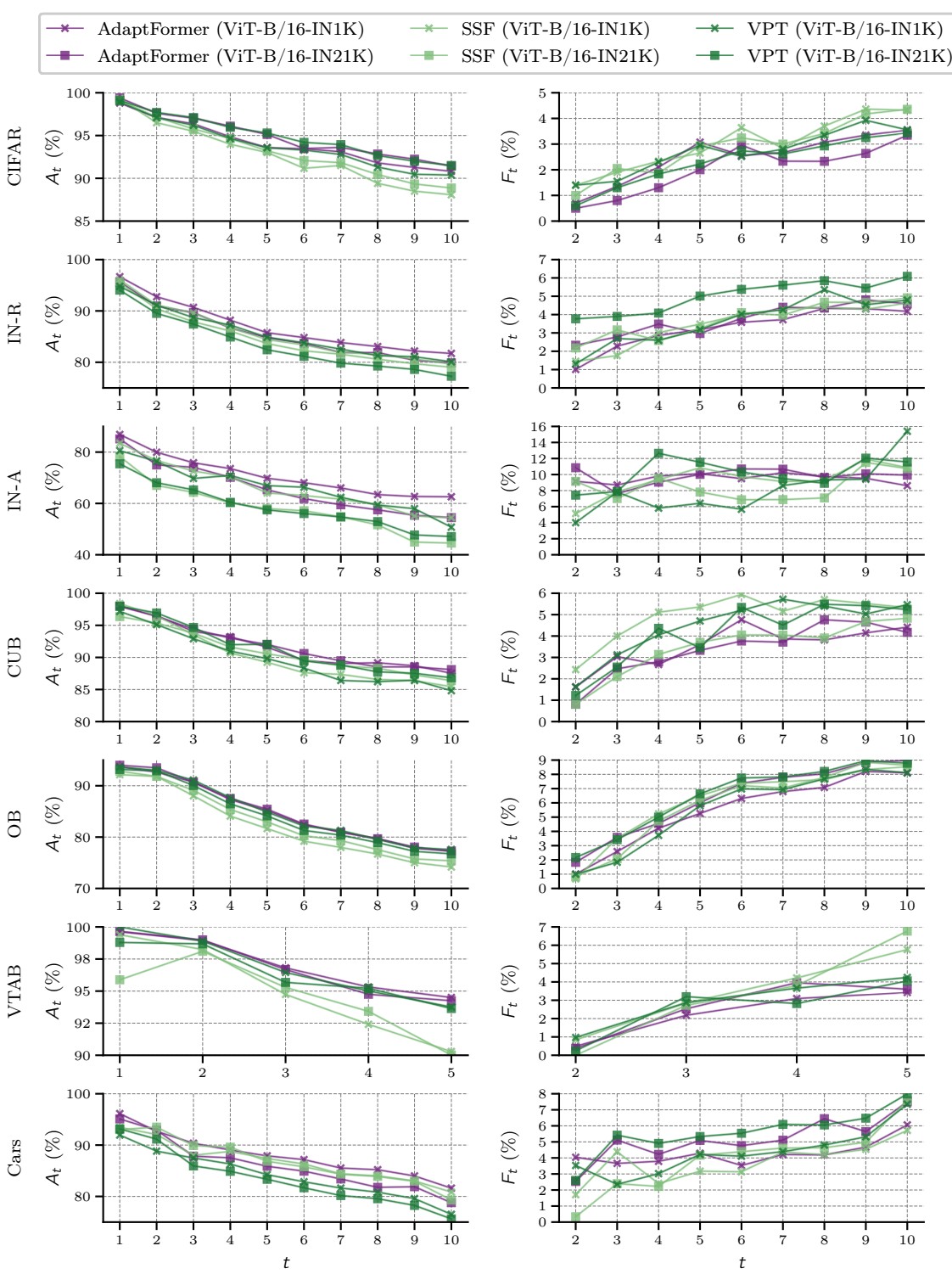

Figure 8: Average accuracy (*left*) and average forgetting (*right*) after training on each task $t$ in the CIL setting: Comparison of different PETL methods (AdaptFormer (Chen et al., 2022), SSF (Lian et al., 2022), and VPT (Jia et al., 2022)) and ViT-B/16 models.

