# OpenReview forum: "Read Between the Layers: Leveraging Multi-Layer Representations for Rehearsal-Free Continual Learning with Pre-Trained Models"
_TMLR — Accepted by TMLR_

### Review · Reviewer_Y7s8 · 2024-05-10

**Summary Of Contributions:**

The paper considers the problem of continual learning (CL) with pre-trained models. The authors study a particular family of CL methods that use prototypes along with parameter-efficient fine-tuning to solve the problem. These methods use the representations from the last layer of the backbone to create prototypes. However, the authors point out that the features present in earlier layers might also be beneficial for the predictions and use this observation to create the LayUP approach. The authors show that this method works well, often outperforming prior state-of-the-art results. They also propose various ablation studies, showing how changing the number of layers impacts the results and how different approaches to adaptation perform.

**Audience:**

Yes

**Claims And Evidence:**

Yes

**Requested Changes:**

I would appreciate additional experiments on using more layers in combination with different methods (RanPAC, Adam). Even a small-scale study on a single dataset would be interesting.

**Strengths And Weaknesses:**

Essentially, the paper proposes a very simple and rather iterative improvement over previous methods such as Adam and NMC+FSA, namely, using a larger number of layers. However, the proposed change does lead to significantly better results. As such, I think this study would be of interest to practitioners and researchers in the area of continual learning with large pre-trained models.

Strengths:

- The proposed improvement is extremely simple and leads to very good results on a wide array of benchmarks.
- The empirical setup is sufficient to support the claims made in the paper. The authors test quite a few datasets and provide useful ablation studies on top of that. Also, I don’t recall any relevant methods that the authors should add to their comparison. One thing to improve would be to add other domains than images, but the current scope is sufficient.
- The paper is clearly written and easy to read.
- The paper describes the related work well.

Weaknesses:

- Since the proposed improvement (using more layers) is somewhat orthogonal to other components of the CL training, I think it is important to ask how it combines with other methods. What if one were to use both pre-trained and fine-tuned features as in Adam but with multiple layers? Or use the RanPAC’s projections using a larger number of layers. I think this is an important angle that was not studied in this paper.
- In the end, the paper is not very novel and fairly incremental with respect to previous work. However, novelty is not an important criterion for TMLR, so I still think it should be accepted, as the empirical gains obtained through this simple improvement should be of interest to the audience.

---

> ### Author Response · Authors · 2024-05-21
> **Response to reviewer Y7s8**
>
> We thank reviewer **Y7s8** for their valuable feedback. We uploaded a revised version of the manuscript with relevant changes highlighted in purple.
>
> ___
>
> **Q1: Additional experiments to answer how LayUP combines with other prototype-based approaches for continual learning, e.g., RanPAC and ADAM.**
>
> **A1:** We have added a new Section 5.7 in the revised version of this manuscript, which contains additional experiments to evaluate the effectiveness of combining RanPAC and ADAM with multi-layer representations from LayUP across multiple datasets and for different values of $k$. We find that integrating LayUP with either class-prototype method consistently improves performance, with the absolute accuracy gain ranging from 0.4% to 31.1%. These results demonstrate that multi-layer representations can successfully enhance existing class-prototype classifiers for continual learning.

---

### Review · Reviewer_nb4p · 2024-05-11

**Summary Of Contributions:**

This paper introduces a class-prototype approach, LayUP, which leverages second-order feature statistics from multiple intermediate layers of a pre-trained network for continual learning (CL). This method addresses the limitation of existing CL approaches that only use features from the final representation layer. The paper claims that LayUP improves performance across various CL benchmarks without requiring rehearsal of prior data.

**Audience:**

Yes

**Claims And Evidence:**

Yes

**Requested Changes:**

1. **Preliminaries Section**: The section could be improved by clarifying the motivation and specific advantages of class-prototype methods over other strategies, especially in terms of handling "full body adaptation" and "linear probing." These terms need better definition and contextualization within the framework of CL.

2. **Definition of $\bar{c}_y$ in Eq. (1)**: The paper should clearly define this term, as it is crucial for understanding the model's operation and the formulation of its prototype-based classification approach.

3. **Choice of $k$ in Eq. (4)**: The method for selecting $k$, which represents the number of layers considered for feature extraction, is not clear. More guidance on how to choose $k$ effectively based on different datasets or task characteristics would be beneficial.

4. **Computational Complexity**: There's a concern that while the inclusion of higher dimension features through $\Phi_{-k}$ might enhance performance, it also leads to increased computational complexity. It would be helpful if the paper could discuss any potential trade-offs and how they might be mitigated.

5. **Consistency Across Datasets**: The observation that LayUP only excels in four out of seven benchmarks might suggest its performance variability with different types of data. This aspect should be explored more thoroughly, perhaps by analyzing characteristics of datasets where LayUP underperforms.

**Strengths And Weaknesses:**

Strengths:

This paper utilizes intra-layer features from pre-trained models, deviating from traditional methods that focus solely on the final layer.

Weaknesses:

1. Does not consistently outperform existing methods across all tested benchmarks, raising questions about its efficacy.

2. Lacks a robust theoretical justification for the use of intra-layer representations and does not fully address potential limitations.

2. The paper does not thoroughly discuss the increased computational complexity and its implications for practical deployment.

---

> ### Author Response · Authors · 2024-05-21
> **Response to reviewer nb4p**
>
> We thank reviewer **nb4p** for their review and appreciate their suggestions. We uploaded a revised version of the manuscript with relevant changes highlighted in purple.
>
> ___
>
> **Q1: Clarification on the motivation and advantages of class-prototype methods, particularly compared with full body adaptation and linear probing.**
>
> **A1:** We have improved the introduction (Section 1) to provide a more detailed explanation of the limitations that class-prototype methods address compared to other strategies for continual learning with pre-trained models. Specifically, we elaborate in more detail on the challenges associated with full-body adaptation and parameter-efficient transfer learning, which necessitates training a linear probe.
>
> ___
>
> **Q2: Formal definition of $\bar{c}_y$.**
>
> **A2:** We have added a new Equation (1), which provides a formal definition of the term $\bar{c}_y$. Additionally, we have expanded on the context of this term in greater detail in Subsections 3.1 and 3.2.
>
> ___
>
> **Q3: Elaboration on how to choose $k$ based on characteristics of the input data.**
>
> **A3:** We have significantly expanded Subsection 5.5 and added a new Figure 4 to determine whether the choice of $k$ can be informed by prior knowledge about characteristics of the CL datasets. For this purpose, we have plotted the degree of domain gap (measured by maximum mean discrepancy between pre-training and downstream datasets), the intra-class similarity and the inter-class similarity of each dataset against the optimal choice of $k$ with respect to performance. We find that
> - datasets for fine-grained classification of natural images that exhibit high intra-class and inter-class similarity (e.g., Cars and CUB) benefit from a high value of $k$,
> - datasets that are composed of multiple specialized natural-image datasets of very distinct domains that exhibit medium intra-class similarity and low inter-class similarity (e.g., VTAB, OB, or CIFAR) benefit from a medium-to-high value of $k$, and
> -  datasets that consist of atypical or stylized image examples with low intra-class and inter-class similarity (e.g., IN-A or IN-R) benefit from a medium value of $k$.
>
> We additionally find that the degree of domain gap alone does not serve as a good indicator for which value of $k$ maximizes performance.
>
> ___
>
> **Q4: Discussion on the trade-offs beetween multi-layer representation depth and computational complexity.**
>
> **A4:** We added a more detailed discussion about how to mitigate the trade-offs between performance and resource requirements based on the choice of $k$ in Section 5.5.
>
> While adding multi-layer representations ($k > 1$) is generally beneficial, performance gains from adding multi-layer representations generally diminish with increasing $k$. As can be seen in Table 4, the average accuracy gain from choosing $k=6$ over $k=1$ is 2.1%, while choosing $k=12$ over $k=6$ only adds 0.4% accuracy on average. However, the computational complexity during inference, driven by the Gram matrix inversion, scales cubically with $k$ (cf. Section 4.3), resulting in an eightfold increase in complexity for $k=12$ compared to $k=6$. This observation corroborates our analysis in Section 4.1 and Figure 3, indicating that $k=6$ substantially enhances performance while maintaining significantly lower computational and memory costs compared with $k=12$ (and compared with other prototype-based methods for CL, as explained in Section 4.3).
>
> ___
>
> **Q5: Clarification on how different characteristics of the datasets influence the success of LayUP.**
>
> **A5:** LayUP demonstrates superior performance compared with other baselines on datasets with relatively small training data and a substantial domain gap from the ImageNet source domain of the ViT models. This underscores that the advantages of multi-layer representations are particularly significant in low-data regimes and for datasets with statistical properties markedly different from the pre-training domain. Conversely, we can infer that the benefits of multi-layer representations are less pronounced when there is a large amount of data available with only a small to medium domain shift from the pre-training domain. We have added more detailed explanations to Section 5.2, accordingly.

---

### Review · Reviewer_VRkt · 2024-05-14

**Summary Of Contributions:**

This paper proposed a simple approach by using the features from middle layers for rehearsal free continuous learning with a pretrained model. This work improved previous work RanPAC by using features from multiple layers without random projection to achieve better results with less computations.
The experimental results outperformed all prototype-based methods in CIL, DIL and online learning.

**Audience:**

Yes

**Claims And Evidence:**

Yes

**Requested Changes:**

1. I wonder what if the authors combine RanPAC with multi-layer features as the authors did? It seems that the random projection helps a lot (Based on the results in table 1 and 2, when k equals to 1, it is usually worse than RanPAC). I wondering that how good the results will be if combining them?

2. Is it necessary to have consecutive layers, and can the proposed method to be used in the network whose dimension is different from each layer?

3. How to decide the k in practice? As shown in Table 4, the performance varied case by case. Especially when a deep model is using, there are more options of k are available.

4. In algorithm 1, G and cy need to be initialized.

**Strengths And Weaknesses:**

Strengths

1. The paper is well-written and east to understand.
2. The idea is simple and effective, without complicated processes to achieve good results
3. The flow of paper is smooth, from the inspiration of feature importance of middle layers and then design how to use the features together to help performance.


Weaknesses
1. Exploiting intra-layer features via GRAM matrix is explored in RANPAC, this paper extends that to include middle layers.
2. In contribution 2, the authors mentioned to decorrelate class prototypes by both intra- and inter-layer features but for the remaining paper, the authors do not discuss the inter part.

---

> ### Author Response · Authors · 2024-05-21
> **Response to reviewer VRkt**
>
> We appreciate reviewer **VRkt**'s feedback and suggestions. We uploaded a revised version of the manuscript with relevant changes highlighted in purple.
>
> ___
>
> **Q1: Discussion of the "inter-layer" features in addition to "intra-layer" features, as mentioned in the contributions.**
>
> **A1:** "[...] leveraging cross-correlations between intra-layer and inter-layer features [...]" means that since features of multiple layers are combined via concatenation to construct prototypes for LayUP, the Gram matrix inversion as per Equation (6) captures cross-correlations from features **within and between** layers. We find in Section 4.1 that this strategy is superior to averaging across multiple layer-specific classifiers that only capture cross-correlations **within** each layer.
>
> However, we acknowledge the notion of "intra-layer" and "inter-layer" features to be confusing, which is why we have edited the description of the contribution in Section 1 for better clarification. We additionally have replaced the notion of "intra-layer" features by "multi-layer" features in the paper, to emphasize the core of our idea to extract features from multiple layers for prototype construction.
> ___
>
> **Q2: Additional experiments to answer how LayUP combines with RanPAC.**
>
> **A2:** We have conducted additional experiments with combinations RanPAC and ADAM with intra-layer representations from LayUP (see also reply to reviewer **Y7s8**), and find that integrating LayUP with other class-prototype methods for continual learning consistently improves performance across all datasets, as detailed in the updated Section 5.7 and its corresponding tables.
>
> ___
>
> **Q3: Questions about (1) whether it is necessary to use features from consecutive layers and (2) whether LayUP works for models with different layer output dimensions.**
>
> **A3:**
> (1) While LayUP can be applied with concatenated features of any subset of layers, empirically finding the best combination of layer features for LayUP among the power-set of layer combinations is very expensive. At the same time, Figure 2 indicates that the quality of the added information of each layer for classification strictly decreases with network depth, which suggests that using the embeddings of the last consecutive layers for prototype construction adds the most benefit to the LayUP method.
>
> (2) Mathematically, our proposed method can be applied to networks with different dimensions per layer. We do not anticipate different results for LayUP in this scenario, as our results in Section 5.5 demonstrate its high robustness and stabilized performance across different values of $k$, which effectively translates to varying prototype dimensionalities.
>
> ___
>
>
> **Q4: Elaboration on how to choose $k$ in practice.**
>
> **A4:** We have significantly expanded Subsection 5.5 and added a new Figure 4 to determine whether the choice of $k$ can be informed by prior knowledge about characteristics of the CL datasets. For this purpose, we have plotted the degree of domain gap (measured by maximum mean discrepancy between pre-training and downstream datasets), the intra-class similarity and the inter-class similarity of each dataset against the optimal choice of $k$ with respect to performance. We find that
> - datasets for fine-grained classification of natural images that exhibit high intra-class and inter-class similarity (e.g., Cars and CUB) benefit from a high value of $k$,
> - datasets that are composed of multiple specialized natural-image datasets of very distinct domains that exhibit medium intra-class similarity and low inter-class similarity (e.g., VTAB, OB, or CIFAR) benefit from a medium-to-high value of $k$, and
> -  datasets that consist of atypical or stylized image examples with low intra-class and inter-class similarity (e.g., IN-A or IN-R) benefit from a medium value of $k$.
>
> We additionally find that the degree of domain gap alone does not serve as a good indicator for which value of $k$ maximizes performance.
>
> ___
>
>
> **Q5: Formal initialization of $G$ and $c_y$.**
>
> **A5:** We have updated Algorithm 1 to include the initialization of $G$ and $c_y$ for clarity.

---

### Decision · Action_Editor_woFY · 2024-06-19

**Recommendation:** Accept as is

**Comment:**

The paper's provides a valuable baseline for the continual learning,  grounded in a simple but efficient approach to leverage feature statistics from intermediate layers of a pre-trained network for rehearsal-free continual learning.

The revised version of the paper thoroughly addressed the reviewers' concerns around a number of criteria and would be a good addition to the journal.  As described in the justification for the audience, the authors address a number of points of concern. Therefore, the recommendation is to accept the work.

The paper does not provide a survey, and its purpose is not to reproduce other methods. So, I would suggest there would be no recommendations for certifications in this case.

**Audience:**

I believe the paper has audience, and all the reviewers are also coming out in support of this statement.

Reviewer  Y7s8 notes that the proposed method can be seen as an iterative improvement over prior work, but at the same time, its performance is quite strong, and it outperforms many of the previous, more sophisticated methods. As such, I think the insights provided here would be valuable for the community.

Reviewer   VRkt notes that this method established a new baseline for rehearsal-free continuous learning.

Reviewer nb4p believes that the response addresses the concerns.

**Claims And Evidence:**

The authors have prepared a thorough response in the rebuttal, addressing the concerns from the reviewers.

Namely, the authors addressed the concerns around:
- how to choose $k$ in practice  (Reviewer nb4p, Reviewer VRkt)
- consistency of performance across datasets (Reviewer nb4p)
- discussion whether it is necessary to use consecutive layers (Reviewer VRkt)
- improvement of clarity of preliminary sections (Reviewer nb4p)
- discussion about combination with  RanPAC (Reviewer Y7s8, VRkt)